# Richness for Tumor-Infiltrating B-Cells in the Oral Cancer Tumor Microenvironment Is a Prognostic Factor in Early-Stage Disease and Improves Outcome in Advanced-Stage Disease

**DOI:** 10.3390/cancers17010113

**Published:** 2025-01-01

**Authors:** Irene H. Nauta, Dennis N. L. M. Nijenhuis, Sonja H. Ganzevles, Pamela I. Raaff, Jan Kloosterman, Elisabeth Bloemena, Ruud H. Brakenhoff, C. René Leemans, Rieneke van de Ven

**Affiliations:** 1Department of Otolaryngology/Head and Neck Surgery, Vrije Universiteit, Amsterdam UMC, Boelelaan 1117, P.O. Box 7057, 1007 MB Amsterdam, The Netherlands; i.nauta@amsterdamumc.nl (I.H.N.); d.n.l.m.nijenhuis@amsterdamumc.nl (D.N.L.M.N.); s.ganzevles@amsterdamumc.nl (S.H.G.); p.raaff@amsterdamumc.nl (P.I.R.); j.kloosterman@amsterdamumc.nl (J.K.); rh.brakenhoff@amsterdamumc.nl (R.H.B.); cr.leemans@amsterdamumc.nl (C.R.L.); 2Cancer Biology and Immunology, Cancer Center Amsterdam (CCA), 1081 HV Amsterdam, The Netherlands; e.bloemena@amsterdamumc.nl; 3Tumor Immunology Theme, Amsterdam Institute for Immunity and Infectious Diseases, Amsterdam, The Netherlands; 4Department of Maxillofacial Surgery/Oral Pathology, Academic Medical Centre for Dentistry, Amsterdam, The Netherlands; 5Department of Pathology, Vrije Universiteit, Amsterdam UMC, De Boelelaan 1117, 1007 MB Amsterdam, The Netherlands

**Keywords:** oral cavity squamous cell carcinoma, tumor-infiltration lymphocytes, B-lymphocytes, prognosis

## Abstract

This study demonstrates that oral cavity squamous cell carcinomas with high levels of tumor-infiltrating B-lymphocytes (TIL-Bs) have a more favorable prognosis, irrespective of the total amount of tumor-infiltrating lymphocytes. Particularly in early-stage disease, the amount of TIL-Bs could be used to stratify patients who do well on standard-of-care therapy and patients who might need treatment intensification. Further research is warranted to unravel the potential cross-talk between and prognostic value of TIL-Bs and T-lymphocytes combined and their relevance in response to immunotherapy treatments, as well as identify the different subpopulations of TIL-Bs with regard to prognosis.

## 1. Introduction

Oral cavity squamous cell carcinoma (OCSCC) is a form of head and neck squamous cell carcinoma (HNSCC) arising in the mucosal linings of the lips, cheeks, floor of the mouth, mobile tongue, gingiva, hard palate, or retromolar trigone [1]. Similar to other HNSCC sites, OCSCC generally results from long-term exposure to tobacco and alcohol [2]. Although the oral cavity lies adjacent to the oropharynx, tumors in the oral cavity are rarely caused by high-risk human papillomavirus (HPV) infection [3,4]. Despite intensive, frequently multimodal treatment, five-year overall survival rates of OCSCC patients do not exceed 65% [5,6], and disease recurrence is seen in half of the patients, so improvement in prognosis is highly desired.

One of the mechanisms that may influence OCSCC outcome is the activation of the host immune system. For HNSCC, several studies indicated a positive prognostic role of tumor-infiltrating T-lymphocytes and regulatory T-cells (Tregs) [7,8,9,10]. In OCSCC specifically, high numbers of CD8+, CD45RO+, and CD57+ tumor-infiltrating T-lymphocytes were associated with improved survival [11].

While the role of tumor-infiltrating T-cells has been studied extensively, the prognostic impact of tumor-infiltrating B-lymphocytes (TIL-Bs) in OCSCC remains somewhat ambiguous. Two studies by Phanthunane et al. demonstrated a prognostic benefit of both CD20+ TIL-B niches and CD20+ TIL-B-cell clusters at the invasive margin in early-stage oral tongue carcinoma [12,13]. When considering other HNSCC subsites, most studies report a favorable effect of B-cells on HNSCC patient survival [14,15,16,17], although one study on 98 stage II-IV laryngeal carcinomas found that tumors with high levels of CD20+ TIL-Bs were more likely to develop distant metastases [18].

We hypothesized that the presence of TIL-Bs in OCSCC is prognostically beneficial. The primary aim of the current study was therefore to better understand the role of TIL-Bs in OCSCC. We performed a quantitative analysis of tumor-infiltrating lymphocytes, emphasizing B-cells, in a consecutive cohort of both early- and advanced-stage OCSCC patients. We correlated the percentage of TIL-Bs with the total amount of TILs and with OCSCC prognosis. Secondly, we investigated subgroups of twelve TIL-B-rich and TIL-B-poor tumors that were stained for CD3 and CD8 to determine differences in T-cell infiltration. At last, further spatial interaction between T- and B-cells was evaluated in six samples.

## 2. Materials and Methods

### 2.1. Study Population

Patients were included from a consecutive cohort of TNM-8 stage I-IVB OCSCC patients treated with curative intent at Amsterdam UMC, location VUmc, between 2008 and 2014 (N = 386). This study was conducted in agreement with the Declaration of Helsinki and the medical ethical guidelines in the Code of Conduct for Proper Secondary Use of Human Tissue of the Dutch Federation of Biomedical Scientific Societies. The Institutional Review Board of VUmc approved the use of the material for immune profiling within this study as a non-WMO protocol (2021-0511).

As B-lymphocytes are especially found in the invasive margin [19], it was considered necessary that all patients should have treatment-naïve formalin-fixed, paraffin-embedded (FFPE) tumor tissue available from either the primary tumor resection or large diagnostic excisional biopsies in which the tumor was surrounded by healthy tissue (as assessed by the corresponding diagnostic H & E-stained sections by a pathologist). Unfortunately, it was technically not feasible to section, stain, and analyze each FFPE block for each tumor, since some resection specimens are made up of dozens of FFPE blocks. We therefore decided to select the one FFPE block from the excision specimen that contained the largest tumor area plus a cuff of tumor-free stroma on the corresponding HE slide. Since we did this uniformly for each tumor, we believe that the selected FFPE blocks are equally representative of the corresponding tumor. An example is included in Appendix A. Of note, verrucous carcinomas, ulcerating tumors, and tumors infiltrating salivary glands were excluded, since ulcers and salivary glands are often enriched for lymphocytes. This left us with 222 evaluable OCSCCs.

### 2.2. Immunohistochemistry

Immunohistochemistry was performed as described previously [20], with minor adaptations: antigen retrieval was performed using 0.01 M sodium citrate (pH 6.0) for CD45 and Tris/EDTA (pH 9.0) for CD19. The slides were blocked with PBS with 10% normal goat serum (NGS) and 1% bovine serum albumin (BSA) for 30 min prior to incubation with isotype control, CD45 (clone: 2B11, Dako (Glostrup, Denmark)) or CD19 (clone: LE-CD19, Dako) specific antibodies for one hour at room temperature (RT) (CD45) or overnight at 4 °C (CD19) in PBS supplemented with 1% BSA and 2% NGS. For T-lymphocyte quantification in TIL-B-rich vs. TIL-B-poor tumor cases, twelve TIL-B-rich and twelve TIL-B-poor tumors were matched on pT-stage and tumor growth patterns. Consecutive sections were prepared from the same tumor block that was used for the TIL-B quantification. Sections were stained for CD3 (clone: A0452, Dako) and for CD8 (clone: C8/14–48, Dako) and respective isotype control antibodies (clone: X0903 and clone: X0931, both Dako) using the IHC protocol described for CD19.

### 2.3. Quantification of Tumor-Infiltrating Lymphocytes

All stained sections were digitally scanned with the Vectra^®^ Polaris™ Imaging System (Akoya Biosciences, Inc. (Marlborough, MA, USA)) at the Cytometry and Microscopy Core Facility of the Cancer Center Amsterdam and analyzed using Fiji software (version 1.53t) by ImageJ. With this software, the total tumor area on each section was manually selected (an example is included in Appendix A), and the percentages of CD45+ and CD19+ immune cells and CD3+ and CD8+ T-lymphocytes were calculated, covering the tumor area as the denominator (consisting of the tumor itself, including stromal areas, and a cuff of tumor-free stroma at the tumor-invasive border). White areas present within the tumor area were subtracted by the software from the total calculated tumor area.

We performed formal cut-off point analyses to determine the optimal cut-off point for OCSCC for OS for CD19^+^ TIL-Bs. For CD19, the ideal cut-off point would be 0.14% (Appendix A, black vertical cutpoint line). However, this would leave us with very few TIL-B-poor cases, making it impossible to perform reliable statistics. To ensure that we had enough cases in both the TIL-B-poor group and the TIL-B-rich group for reliable statistics, we decided to divide the groups based on the median. For CD19+ TIL-B, the median percentage was 0.90%, which fell nicely after the first peak within the distribution in the cut-off analyses, and within the high range of the statistical rank (Appendix A, blue vertical line). For the %CD45+ lymphocytes and the CD19/CD45 ratio, the median was also chosen.

### 2.4. Multiplexed Fluorescent Immunohistochemistry (m-fIHC)

Staining was performed as recently described [21]. In short, 3 µm FFPE sections were deparaffinized by incubation at 60 °C for 1 h, followed by 2 times 7.5 min in xylene. After 5 min incubations in 100% and 96% ethanol, endogenous peroxidases were blocked by 0.3% H_2_O_2_ in methanol for 20 min. All heat-induced epitope retrieval (HIER) steps were performed with Tris/EDTA buffer pH9.

The Opal 6-plex Manual Detection Kit (Akoya, NEL861001KT) was used for the m-fIHC staining, according to the manufacturer’s protocol. For CD44v6 (Vumc (Amsterdam, The Netherlands), clone U36, 1:200), CD3 (Dako A0452, 1:250), CD8 (Dako M7103, 1:250), and FoxP3 (Abcam (Cambridge, UK) Ab20034, 1:100) staining, slides underwent similar staining rounds. Aspecific binding was blocked by Antibody Diluent/Block (Akoya) solution for 10 min. Primary antibodies were incubated for 1 h at RT followed by 15 min goat-anti-rabbit/mouse-HRP (Akoya). Opal signal was generated by applying TSA-fluorochromes Opal690, Opal620, Opal480, and Opal520 (1:100), for 10 min. Antibody complexes were stripped by HIER. For CD19 (Abcam Ab134114, 1:100), aspecific binding was blocked by 20% NGS in Antibody Diluent/Block for 30 min. After overnight incubation with the primary antibody, biotin-labeled goat-anti-mouse (Invitrogen (Carlsbad, CA, USA) 31800, 1:300) was applied, followed by Streptavidin-HRP, both for 30 min at RT. CD163 (Novubio, Minneapolis, Minnesota, USA, NB110-59935, 1:100) was incubated overnight, and Opal780 staining was performed according to the manufacturer’s protocol. Cell nuclei were visualized with DAPI, and slides were mounted in Prolong Diamond Antifade Mount (Invitrogen P36970).

Slides were imaged using PerkinElmer’s Vectra Polaris, spectrally unmixed with InForm software (version 2.6). Cell segmentation and the annotation of the tumor area were performed using QuPath (version 0.4.3). Spatial analysis was performed using the SPIAT package (version 1.2.2). One TIL-B-poor sample was excluded from the analysis as it failed quality assessment criteria.

### 2.5. Clinical Endpoints

Clinical endpoints included five-year overall survival (OS) and disease-free survival (DFS). OS was defined as the time between histologically confirmed OCSCC diagnosis and death from any cause. DFS was defined as the time between histologically confirmed OCSCC diagnosis and disease recurrence or death from any cause. Disease recurrence comprised local recurrence (within 2 cm and within 3 years of diagnosis of the tumor under investigation), regional recurrence (lymph node metastases in previously treated levels of the neck), distant metastases, and second primary HNSCC (occurring more than 2 cm from and more than 3 years after the index tumor). Patients with residual disease or occult lymph node metastases were censored at the respective dates.

### 2.6. Statistical Analyses

Statistical analyses were performed using IBM^®^ SPSS^®^ Statistics 26 or Graphpad Prism version 8. *p*-values < 0.05 were considered statistically significant.

## 3. Results

### 3.1. Demographic and Clinical Characteristics

Demographic and clinical characteristics of the 222 OCSCC patients included in this study are depicted in Table 1, left column. The mean age of the cohort was 63 years, 56.3% of patients were male, and 74.3% of patients had little to no comorbidity. More than half of the tumors were located in the mobile tongue; 80.2% were classified as cT1–2 tumors, and 22.5% had lymph node metastases at the time of diagnosis.

The median percentages of CD45+ TILs, CD19+ TIL-Bs, and the CD19+/CD45+ ratio were 11.90%, 0.90%, and 9.01%, respectively. Based on these median percentages, OCSCCs were classified as “TIL-rich” (CD45+ TILs > 11.90%) or “TIL-poor” (CD45+ TILs ≤ 11.90%); “TIL-B-rich” (CD19+ TIL-Bs > 0.90%, Figure 1A) or “TIL-B-poor” (CD19+ TIL-Bs ≤ 0.90%, Figure 1B); and “CD19/CD45 ratio-high” (CD19+/CD45+ ratio > 9.01%) or “CD19/CD45 ratio-low” (CD19+/CD45+ ratio ≤ 9.01%).

Since this study focused primarily on TIL-Bs, demographic and clinical characteristics of the “TIL-B-rich” and “TIL-B-poor” OCSCCs are included in Table 1, together with their corresponding *p*-values. Of note, separate tables with demographic and clinical characteristics for pathological disease stage I–II OCSCC patients and pathological disease stage III–IV OCSCC patients can be found in the Appendix A, respectively).

Apart from tumor location, cN-stage, and treatment regimen, no differences in demographic and clinical characteristics between TIL-B-rich and TIL-B-poor OCSCCs were observed. Compared to TIL-B-poor OCSCCs, TIL-B-rich OCSCCs were significantly more often located in the mobile tongue (69.8% vs. 45.8%, *p* = 0.001), generally did not display lymph node metastases at diagnosis (86.2% vs. 69.0%, *p* = 0.021), and were less likely to have received postoperative (chemo)radiotherapy (23.9% vs. 41.6%, *p* = 0.031). When differentiating between pathological disease stage I–II OCSCC (early stage, N = 100) and pathological disease stage III–IV OCSCC (advanced stage, N = 114), there were no significant differences in demographic or clinical parameters between the early-stage TIL-B-rich (N = 59) and TIL-B-poor (N = 41) groups (Appendix A). Within the pathological disease stage III–IV patients, TIL-B-rich patients (N = 45) had significantly fewer unit years (*p* = 0.012) and a lower clinical disease stage (*p* = 0.024) compared to the TIL-B-poor group (N = 69) (Appendix A). Of note, 8 of the 222 OCSCC patients included were treated non-surgically, and hence, no pathological disease stage was assigned to these patients.

### 3.2. Pathological Characteristics

In Table 2, pathological characteristics are displayed both for the complete OCSCC cohort and for TIL-B-rich and TIL-B-poor OCSCC separately. Mean tumor diameter, tumor depth of invasion, and pT-stage were significantly higher in TIL-B-poor OCSCC than in TIL-B-rich tumors (*p* = 0.001, *p* = 0.005, and *p* = 0.002, respectively). TIL-B-rich OCSCCs more frequently exhibited a cohesive growth pattern (47.7% vs. 34.5%, *p* = 0.016) and were less likely to have perineural invasion (14.7% vs. 28.3%, *p* = 0.032) or lymph node metastases with extracapsular spread (12.8% vs. 27.4%, *p* = 0.026). When differentiating between early-stage OCSCC and advanced-stage III–IV OCSCC, there were no significant differences in histopathological parameters between the advanced-stage TIL-B-rich and TIL-B-poor groups (Appendix A). Within the early-stage OCSCC patients, TIL-B-rich OCSCC more often exhibited a cohesive growth pattern than the TIL-B-poor group (*p* = 0.013, Appendix A).

### 3.3. Survival of TIL-B-Rich and TIL-B-Poor OCSCCs

In the cohort studied, five-year OS was 75.0% for TIL-B-rich OCSCCs and 54.2% for TIL-B-poor OCSCCs (*p* < 0.001, Figure 2A), while five-year DFS was 66.9% and 50.9%, respectively (*p* = 0.003, Appendix A). As pT-stage was significantly unequally distributed between TIL-B-rich and TIL-B-poor OCSCCs, and extracapsular spread was relatively more frequently observed among TIL-B-poor OCSCCs (Table 2), we also generated separate five-year overall survival curves for TNM-8 disease stage I–II (Figure 2B) and disease stage III–IV (Figure 2C), as well as five-year disease-free survival curves for TNM-8 disease stage I–II (Appendix A) and disease stage III–IV (Appendix A). TIL-B richness remained a prognostic determinant in disease stage I–II OCSCC for both 5-year OS (*p* < 0.001) and DFS (*p* < 0.026). In the advanced-stage patients, there was no significant difference in 5-year OS (*p* = 0.212) or DFS (*p* = 0.190) between TIL-B-rich and TIL-B-poor cases.

Since the presence and prognostic impact of TIL-Bs may be biased by the total amount of tumor-infiltrating lymphocytes, we subsequently determined the five-year OS of the CD19/CD45 ratio (Figure 3). For OCSCCs with a high CD19/CD45 ratio, the five-year OS was 72.0%, while OCSCCs with a low CD19/CD45 ratio had a five-year OS of 57.1% (*p* = 0.008).

Considering that TIL-B-rich and TIL-B-poor OCSCC had distinct histopathological characteristics that may influence outcome (Table 2), we performed a multivariate Cox regression analysis, including tumor diameter, tumor depth of invasion, tumor pattern of invasion, perineural invasion, differentiation grade, pT-stage, extracapsular spread, and treatment regimen as model covariates. After correction for these variables, the percentage of TIL-Bs remained a significant prognostic factor for five-year OS (adjusted hazard ratio = 1.9; 95% confidence interval: 1.1–3.6; *p* = 0.033). When the same multivariate Cox regression was applied separately on stage I–II and stage III–IV OCSCC, the prognostic effect remained significant in the early-stage group but not in the advanced-stage group (stage I–II: adjusted hazard ratio = 8.5, 95% confidence interval: 1.5–46.6, *p* = 0.014; stage III–IV: adjusted hazard ratio = 1.3, 95% confidence interval: 0.7–2.6; *p* = 0.458).

### 3.4. T-Lymphocytes in TIL-B-Rich vs. TIL-B-Poor OCSCC

Given the known literature on the positive prognostic role of CD8+ cytotoxic T-lymphocytes in tumors, including reports for OCSCC, we were curious whether TIL-B-rich tumors in our cohort also contained higher levels of T-lymphocytes. For this purpose, TIL-B-rich and TIL-B-poor tumors were matched based on pT-stage, focusing on pT1 and pT2 tumors, and tumor pattern of invasion (cohesive vs. non-cohesive). CD3+ and CD8+ T-lymphocytes were quantified in twelve matched pairs (TIL-B-rich vs. TIL-B-poor). While CD8+ cytotoxic T-lymphocytes within the tumor area were comparable between the TIL-B-rich and TIL-B-poor tumors (Figure 4A), significantly more CD3+ T-lymphocytes were present in TIL-B-rich tumors (*p* = 0.007) (Figure 4B), suggesting a putative cross-talk between TIL-B and CD4+ (CD3+/CD8−) T-lymphocytes within the tumor microenvironment of OCSCC.

### 3.5. Spatial Characterization of TIL-Bs in the Tumor Microenvironment of OCSCC

To spatially characterize TIL-B infiltration and interaction with other immune subsets, we compared five TIL-B-poor and six TIL-B-rich OCSCC using an m-fIHC panel optimized for tumor cells, CD163+ macrophages, CD4+, CD8+, and FoxP3+ regulatory T-lymphocytes (Figure 5A and Figure 6A,B).

Although it had a small sample size, our m-fIHC analysis confirmed the significant increase in CD4+ T-lymphocytes (CD3+/CD8−) in the tumor area of TIL-B-rich tumors, while CD8+ T-lymphocyte and CD163+ macrophage infiltration levels were comparable (Figure 5B). Although not significant (*p* = 0.06), we observed a slight increase in Treg infiltration, which indicates that Tregs (CD3+/FoxP3+/CD8−) are part of the increased CD3+ T-lymphocytes seen in our chromogenic IHC data. To assess TIL-B interaction partners, we performed a minimal distance analysis using TIL-Bs as reference cells (Figure 5C). In general, CD4+ T-lymphocytes were closest to TIL-Bs compared to the other immune subtypes and showed the least amount of spread between OCSCC samples. In TIL-B-rich OCSCC, we observed a significant reduction in the TIL-B/CD4+ T-lymphocyte distance compared to TIL-B-poor OCSCC, indicative of increased CD4+ T-lymphocyte/TIL-B co-localization upon B-cell infiltration (Figure 5D). Interestingly, TIL-Bs were significantly closer to tumor cells in TIL-B-poor OCSCC. This could be due to the tendency of TIL-Bs to form clusters near the tumor border in TIL-B-rich OCSCC (Figure 6).

## 4. Discussion

This study demonstrates the prognostic impact of tumor-infiltrating B-lymphocytes in stage I-IV OCSCC. Up until now, studies on the role of TIL-Bs within various HNSCC sites generally focused on oro- and hypopharyngeal cancers [10,14,22,23]. Two studies in OCSCC did not observe a significant association between CD20+ TIL-B and OS in early-stage oral tongue cancer but only found a correlation when combining the presence of CD20+ TIL-B and CD4+ T-cells [12]. Particularly, the proximity of CD20+ TIL-B and T follicular helper cells within the tumor invasive border were associated with a favorable prognosis [13,21]. Of note, advanced-stage oral tongue cancer and other oral cavity subsites were not included in their analyses [12,13].

Our study is one of few that used CD19 as a marker for B-cells, while in most other studies, CD20 was used [7,12,13,14,16,17,18,19,22,24]. Unlike CD20, CD19 expression is maintained in the majority of plasma cells. The percentage of CD19+ TIL-Bs in our cohort of OCSCCs therefore includes the amount of tumor-infiltrating plasma cells. Although the independent prognostic role of plasma cells in HNSCC remains unstudied, and cannot be determined based on the current study, a survival benefit of tumor-infiltrating plasma cells has been described in ovarian cancer and non-small-cell lung cancer [25,26].

The observed survival benefit of high levels of CD19+ TIL-Bs in our OCSCC cohort was more pronounced in stage I–II OCSCCs than in stage III–IV OCSCCs. A paper by Distel et al. reported on a contradictory effect of CD20+ TIL-Bs in oro- and hypopharyngeal squamous cell carcinomas [22]. They found that high percentages of CD20+ TIL-Bs were only beneficial in low-risk patients (i.e., patients with early-stage disease treated with primary surgery and postoperative radiotherapy), but that it had a detrimental effect on high-risk patients (i.e., patients with advanced-stage disease treated with definitive chemoradiotherapy). Although the prognostic benefit of TIL-Bs in our study was only significant in stage I–II OCSCCs and not in stage III–IV OCSCCs, we could not verify the detrimental effect of high levels of B-cells in stage III–IV OCSCCs as described by Distel, as in our study, especially up to 36 months after diagnosis, TIL-B richness also seemed to benefit the patients with advanced-stage disease.

Compared to TIL-B-poor OCSCCs, TIL-B-rich OCSCCs were more likely to have favorable histopathological characteristics such as a cohesive growth pattern. A correlation between the tumor growth pattern and the tumor immune microenvironment has already been reported in gastric and esophageal cancer [27,28]. Multivariate Cox regression analysis including tumor diameter, tumor depth of invasion, tumor pattern of invasion, perineural invasion, differentiation grade, pT-stage, extracapsular spread, and treatment regimen showed that the percentage of TIL-Bs remained a significant prognostic factor for five-year OS in our cohort. We can only speculate that the higher abundance of TIL-Bs has contributed to the more favorable histopathological characteristics of the growing tumors by actively being involved in an anti-tumor immune response. This patient cohort was diagnosed with OCSCC between 2008 and 2014. At that time, the two-category classification of the pattern of invasion (cohesive vs. noncohesive) was a recognized AJCC classification and was used by pathologists. Only recently, the AJCC classification has incorporated the five-category classification (the WPOI). In a recent publication, we showed that in OCSCC, there was a good correlation between WPOI class 1–3 and a cohesive pattern of invasion, and between WPOI class 4–5 and a noncohesive growth pattern [29]. Since a five-category distribution would reduce the statistical power to find differences in this cohort of 109 TIL-B-rich and 113 TIL-B-poor cases, we decided to use the two-category classification.

Our study demonstrates that the prognostic impact of TIL-Bs in OCSCCs is independent of the total number of tumor-infiltrating lymphocytes, as displayed by the more favorable five-year OS and DFS of OCSCCs with a high CD19/CD45 ratio compared to oral cancers with a low CD19/CD45 ratio. Indubitably, the various subsets of lymphocytes do not function independently. Alternately, they interact with each other to initiate a complex series of immune responses. In a selection of TIL-B-rich and TIL-B-poor OCSCCs, which were matched for pT-stage and growth pattern of invasion, we found the presence of CD8+ T-lymphocytes to be similar, despite significantly more CD3+ T-lymphocytes in the TIL-B-rich group. Our exploratory spatial m-fIHC analysis showed that TIL-Bs are mostly co-localized with CD4+ T-lymphocytes, especially in TIL-B-rich OCSCC. These observations were also reported by us [21] in a prospectively collected dataset containing OCSCC and suggest a potential cross-talk between TIL-B and CD4+ T-lymphocytes in OCSCC that could have prognostic value, which is in line with the study by Phanthunane et al. [13]. Importantly, not all TIL-B-rich tumors analyzed contained high frequencies of CD3+ T-lymphocytes, while all TIL-B-poor cases analyzed had low CD3+ T-lymphocyte counts. This might imply that OCSCC could potentially be divided into three groups: OCSCC with high TIL-Bs and high CD3+ TIL levels, a group with high TIL-Bs and low CD3+ TIL levels, and a group with low TIL-Bs and low CD3+ TIL levels. It would require a much larger cohort of patients to unravel the prognostic difference between these groups. Ideally, such analyses would be performed using m-fIHC to take into account the spatial distance between TIL-B and T-lymphocytes and to discriminate regulatory CD4+ T-cells from effector CD4+ T-lymphocytes. Previously, Feng et al. showed the distance between CD8+ cytotoxic T-lymphocytes and Foxp3+ regulatory T-cells within the tumor-invasive border to relate to OS in OCSCC [30].

## 5. Conclusions

We demonstrated that oral cavity squamous cell carcinomas with high levels of CD19+ tumor-infiltrating B-lymphocytes have a more favorable prognosis, irrespective of the total amount of CD45+ tumor-infiltrating lymphocytes, and that, particularly in early-stage disease, this could be used to stratify patients who do well on standard-of-care therapy and patients who might need treatment intensification such as adjuvant radiotherapy or (local) pre-operative immunotherapy regimens. Further research is warranted to unravel the potential cross-talk between and prognostic value of TIL-Bs and CD4+ T-lymphocytes combined and their relevance in response to immunotherapy treatments, as well as identify the different subpopulations of TIL-Bs with regard to prognosis.

## Figures and Tables

**Figure 1 cancers-17-00113-f001:**
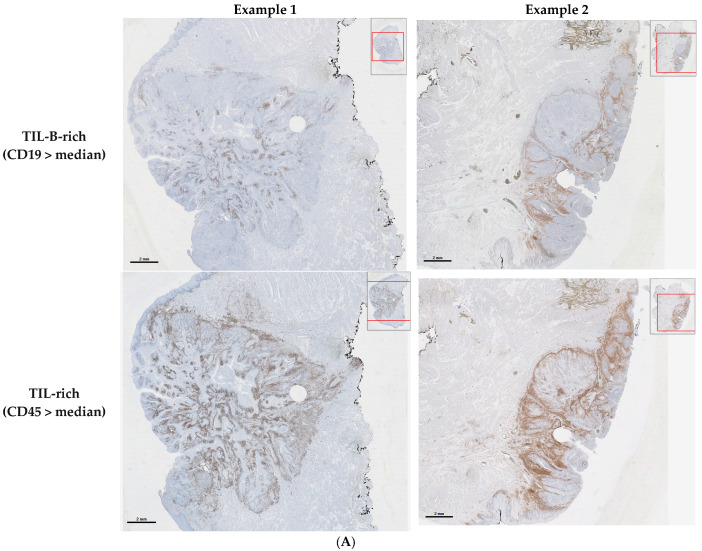
Immunohistochemical analysis of CD45 and CD19 in OCSCC. On the top right of each image, an overview of the slide is shown. The red box identifies the area that has been magnified in the main image. (**A**) Two examples of TIL-rich (CD45 > median) and TIL-B-rich (CD19 > median) OCSCCs. (**B**) Two examples of TIL-rich (CD45 > median) but TIL-B-poor (CD19 ≤ median) OCSCCs. The white circle in the tumor area is caused by a punch biopsy that was taken from the FFPE block for another study. This white area was excluded from the calculated tumor area.

**Figure 2 cancers-17-00113-f002:**
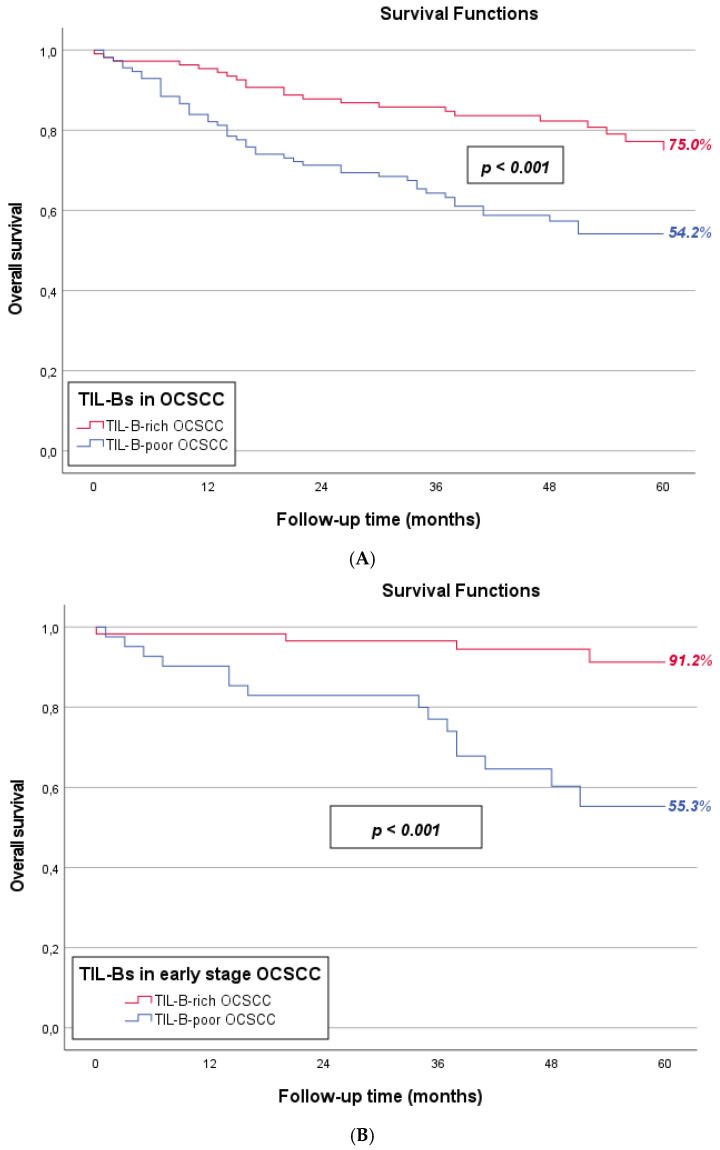
Survival curves of TIL-B-rich (red line) and TIL-B-poor (blue line) OCSCCs. (**A**) Five-year OS was 75.0% for TIL-B-rich OCSCCs and 54.2% for TIL-B-poor OCSCCs (*p* < 0.001). (**B**) Separate survival curves of pathological disease stage I–II (“early-stage disease”) OCSCC. Five-year OS of pathological disease stage I–II was 91.2% for TIL-B-rich OCSCCs and 55.3% for TIL-B-poor OCSCCs (*p* < 0.001). (**C**) Separate survival curves of pathological disease stage III–IV (“advanced-stage disease”) OCSCC. Five-year OS of pathological disease stage III–IV was 57.7% for TIL-B-rich OCSCCs and 52.6% for TIL-B-poor OCSCCs (*p* = 0.212).

**Figure 3 cancers-17-00113-f003:**
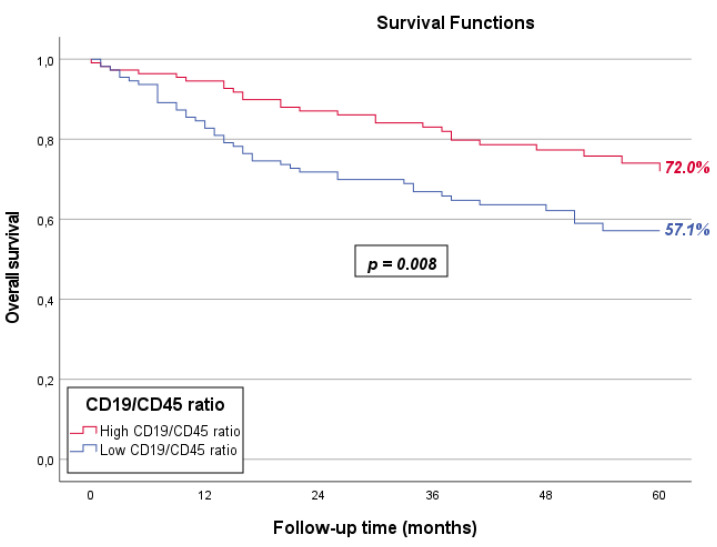
Survival curves of OCSCCs with a high CD19/CD45 ratio (red line) and OCSCCs with a low CD19/CD45 ratio (blue line). Five-year OS was 72.0% for OCSCCs with a high CD19/CD45 ratio and 57.1% for OCSCCs with a low CD19/CD45 ratio (*p* = 0.008).

**Figure 4 cancers-17-00113-f004:**
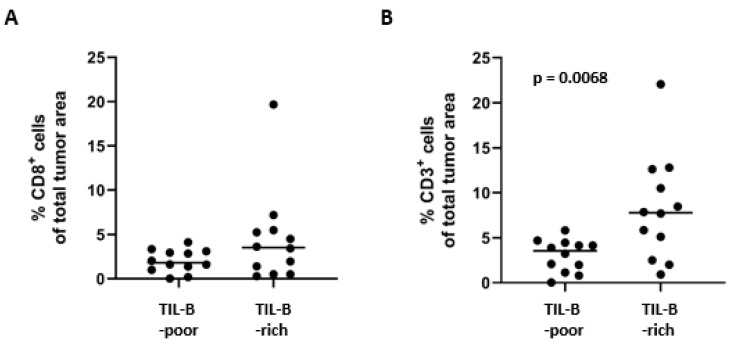
Quantification of CD8+ and CD3+ T-lymphocytes in TIL-B-rich vs. TIL-B-poor OCSCC. (**A**) CD8+ T-lymphocytes and (**B**) CD3+ T-lymphocytes were quantified in 12 TIL-B-poor and 12 TIL-B-rich OCSCCs from the original cohort (*N* = 222), which were matched on pT-stage and growth pattern of invasion. Both cohesive and non-cohesive growing tumors were included. Mann–Whitney tests were performed to determine the statistical significance in T-cell counts between TIL-B-poor and TIL-B-rich OCSCC using Graphpad Prism 8.

**Figure 5 cancers-17-00113-f005:**
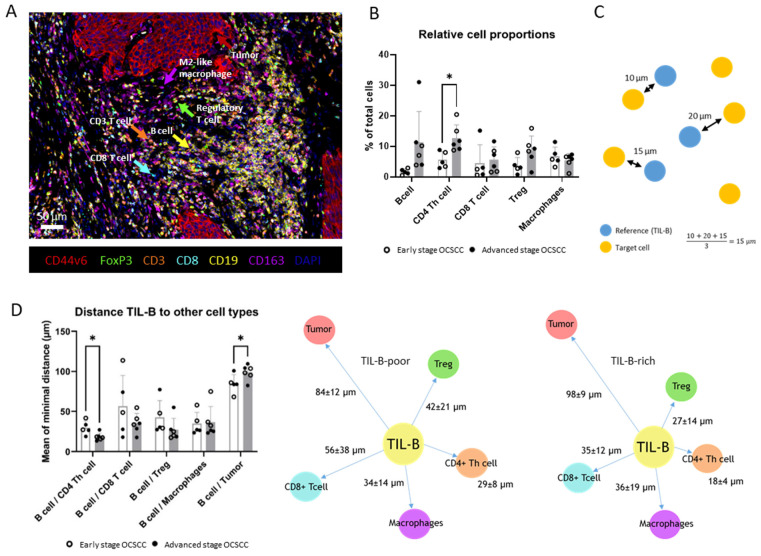
Spatial immune infiltrate analysis of TIL-B-poor and TIL-B-rich OCSCC using mfIHC. Six TIL-B-poor and six TIL-B-rich samples were matched based on the pattern of invasion (cohesive or non-cohesive) and tumor stage (early-stage or advanced-stage). In each group, three early-stage and three advanced-stage OCSCCs were stained and analyzed. One early-stage TIL-B-poor sample was excluded from the analysis due to unreliable staining quality. (**A**) Representation of the mfIHC panel optimized to detect CD44v6^+^ tumor cells, CD163^+^ macrophages, CD19^+^ B-cells, CD3^+^, CD3^+^CD8^+^, CD3^+^CD8^−^ FoxP3^−^ (CD4^+^ helper) T-lymphocytes, and CD3^+^CD8^−^FoxP3^+^Treg cells. (**B**) Cell proportions relative to the total number of detected cells within the tumor area. Open bars represent TIL-B-poor tumors, and closed bars represent TIL-B-rich tumors. The asterisk represents a statistical significant difference between the two appointed categories. (**C**) Spatial analysis was performed by a minimal distance analysis. The shortest distance from each TIL-B to each of the other immune subsets was calculated and corrected for the number of TIL-Bs. (**D**) Mean of the minimal distance between TIL-Bs and the other immune subsets. Open bars represent TIL-B-poor OCSCC, and closed bars represent TIL-B-rich OCSCC. The asterisk represents a statistical significant difference between the two appointed categories. Unpaired *t*-tests were performed between TIL-B-rich and TIL-B-poor conditions using Graphpad Prism 8.

**Figure 6 cancers-17-00113-f006:**
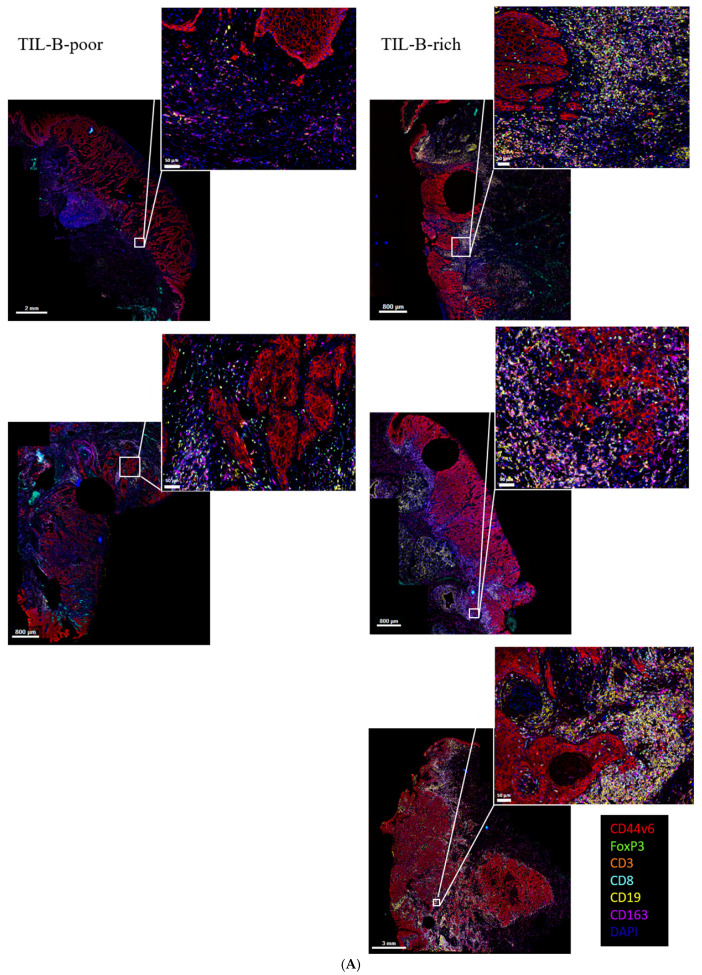
Overview of TIL-B-poor and TIL-B-rich early-stage (**A**) and advanced-stage (**B**) OCSCC mfIHC samples.

**Table 1 cancers-17-00113-t001:** Demographic and clinical characteristics of the studied cohort.

		Total			TIL-B-Rich OCSCC			TIL-B-Poor OCSCC			
		(N = 222)			(N = 109)			(N = 113)			
		Mean (SD)	N	%	Mean (SD)	N	%	Mean (SD)	N	%	*p*-Value *
Age at diagnosis		63 (13)			63 (13)			63 (12)			0.926 ¥
Sex											1.000 †
	Male		125	56.3%		61	56.0%		64	56.6%	
	Female		97	43.7%		48	44.0%		49	43.4%	
Comorbidity (ACE-27)											0.778 †
	None		85	38.3%		45	41.3%		40	35.4%	
	Mild		80	36.0%		39	35.8%		41	36.3%	
	Moderate		41	18.5%		18	16.5%		23	20.4%	
	Severe		16	7.2%		7	6.4%		9	8.0%	
Pack years		25 (23)			23 (24)			26 (22)			0.324 ¥
Unit years		115 (168)			100 (181)			130 (153)			0.190 ¥
Tumor location											**0.001 ‡**
	Mobile tongue		125	56.3%		73	67.0%		52	46.0%	
	Floor of mouth		63	28.4%		28	25.7%		35	31.0%	
	Vestibulum of mouth		16	7.2%		3	2.8%		13	11.5%	
	Hard palate		1	0.5%		1	0.9%		0	0.0%	
	Cheek mucosa		12	5.4%		4	3.7%		8	7.1%	
	Retromolar trigone		5	2.3%		0	0.0%		5	4.4%	
cT-stage (TNM-8)											0.063 †
	T1		107	48.2%		62	56.9%		45	39.8%	
	T2		71	32.0%		30	27.5%		41	36.3%	
	T3		21	9.5%		10	9.2%		11	9.7%	
	T4a		22	9.9%		7	6.4%		15	13.3%	
	T4b		1	0.5%		0	0.0%		1	0.9%	
cN-stage (TNM-8)											**0.021 ‡**
	N0		172	77.5%		94	86.2%		78	69.0%	
	N1		24	10.8%		9	8.3%		15	13.3%	
	N2a		1	0.5%		0	0.0%		1	0.9%	
	N2b		15	6.8%		5	4.6%		10	8.8%	
	N2c		4	1.8%		0	0.0%		4	3.5%	
	N3a		0	0.0%		0	0.0%		0	0.0%	
	N3b		6	2.7%		1	0.9%		5	4.4%	
Disease stage (TNM-8)											**0.002 †**
	Stage I		97	43.7%		58	53.2%		39	34.5%	
	Stage II		50	22.5%		28	25.7%		22	19.5%	
	Stage III		29	13.1%		10	9.2%		19	16.8%	
	Stage IVA		39	17.6%		12	11.0%		27	23.9%	
	Stage IVB		7	3.2%		1	0.9%		6	5.3%	
Type of treatment											**0.031 ‡**
	Surgery		141	63.5%		78	71.6%		63	55.8%	
	Surgery + RT		43	19.4%		16	14.7%		27	23.9%	
	Surgery + CRT		30	13.5%		10	9.2%		20	17.7%	
	Radiotherapy		5	2.3%		3	2.8%		2	1.8%	
	Chemoradiotherapy		2	0.9%		2	1.8%		0	0.0%	
	Bioradiation		1	0.5%		0	0.0%		1	0.9%	
Neck dissection											**0.018 ‡**
	No		106	47.7%		60	55.0%		46	40.7%	
	Unilateral		77	34.7%		36	33.0%		41	36.3%	
	Bilateral		31	14.0%		8	7.3%		23	20.4%	
	n.a.		8	3.6%		5	4.6%		3	2.7%	

* *p*-value between “TIL-B-rich” and “TIL-B-low”. ¥ Student’s *t* test; † Pearson Chi-square test; ‡ Fisher’s exact test. RT = radiotherapy, CRT = chemoradiotherapy (radiotherapy with concomitant cisplatin), bioradiation = radiotherapy with concomitant Cetuximab, n.a. = not applicable.

**Table 2 cancers-17-00113-t002:** Histopathological characteristics of the studied cohort.

		Total			TIL-B-Rich OCSCC			TIL-B-Poor OCSCC			
		(N = 222)			(N = 109)			(N = 113)			
		Mean (SD)	N	%	Mean (SD)	N	%	Mean (SD)	N	%	*p*-Value *
Tumor diameter (cm)		2.02 (1.30)			1.71 (1.20)			2.30 (1.33)			**0.001 ¥**
Tumor depth of invasion (cm)		0.91 (0.94)			0.72 (0.65)			1.09 (1.12)			**0.005 ¥**
Tumor level of invasion											0.067 ‡
	Micro-invasion		7	3.2%		4	3.7%		3	2.7%	
	Lamina propria		99	44.6%		48	44.0%		51	45.1%	
	Submucosa		25	11.3%		14	12.8%		11	9.7%	
	Muscle		61	27.5%		33	30.3%		28	24.8%	
	Bone		20	9.0%		4	3.7%		16	14.2%	
	Skin		2	0.9%		0	0.0%		2	1.8%	
	Could not be assessed		8	3.6%		6	5.5%		2	1.8%	
Tumor pattern of invasion											**0.016 †**
	Cohesive		91	41.0%		52	47.7%		39	34.5%	
	Non-cohesive		106	47.7%		42	38.5%		64	56.6%	
	Could not be assessed		25	11.3%		15	13.8%		10	8.8%	
Lymphovascular invasion											0.156 †
	No		177	79.7%		86	78.9%		91	80.5%	
	Yes		20	9.0%		6	5.5%		14	12.4%	
	Could not be assessed		25	11.3%		17	15.6%		8	7.1%	
Perineural invasion											**0.032 †**
	No		150	67.6%		77	70.6%		73	64.6%	
	Yes		48	21.6%		16	14.7%		32	28.3%	
	Could not be assessed		24	10.8%		16	14.7%		8	7.1%	
Differentiation grade											**0.034 †**
	Well differentiated		33	14.9%		20	18.3%		13	11.5%	
	Moderately differentiated		119	53.6%		59	54.1%		60	53.1%	
	Poorly differentiated		39	17.6%		12	11.0%		27	23.9%	
	Could not be assessed		31	14.0%		18	16.5%		13	11.5%	
pT-stage (TNM-8)											**0.002 †**
	T1		85	38.3%		54	49.5%		31	27.4%	
	T2		60	27.0%		28	25.7%		32	28.3%	
	T3		42	18.9%		14	12.8%		28	24.8%	
	T4a		27	12.2%		8	7.3%		19	16.8%	
	T4b		0	0.0%		0	0.0%		0	0.0%	
	n.a. (no surgery)		8	3.6%		5	4.6%		3	2.7%	
pN-stage (TNM-8)											0.148 ‡
	N0		107	48.2%		59	54.1%		48	42.5%	
	N1		27	12.2%		15	13.8%		12	10.6%	
	N2a		12	5.4%		5	4.6%		7	6.2%	
	N2b		9	4.1%		4	3.7%		5	4.4%	
	N2c		3	1.4%		1	0.9%		2	1.8%	
	N3a		0	0.0%		0	0.0%		0	0.0%	
	N3b		33	14.9%		9	8.3%		24	21.2%	
	n.a. (no neck dissection)		31	14.0%		16	14.7%		15	13.3%	
Surgical margins											0.080 †
	Clear (>5 mm)		159	75.0%		84	81.6%		75	68.8%	
	Close (1–5 mm)		36	17.0%		14	13.6%		22	20.2%	
	Involved (<1 mm)		17	8.0%		5	4.9%		12	11.0%	
Extracapsular spread											**0.026 †**
	No		146	65.8%		79	72.5%		67	59.3%	
	Yes		45	20.3%		14	12.8%		31	27.4%	
	n.a. (no neck dissection)		31	14.0%		16	14.7%		15	13.3%	

* *p*-value between “TIL-B-rich” and “TIL-B-poor”. ¥ Student’s *t* test; † Pearson Chi-square test; ‡ Fisher’s exact test, n.a. = not applicable.

## Data Availability

The data that support the findings of this study are available from the corresponding author upon reasonable request.

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
