# Peer review of "Richness for Tumor-Infiltrating B-Cells in the Oral Cancer Tumor Microenvironment Is a Prognostic Factor in Early-Stage Disease and Improves Outcome in Advanced-Stage Disease"

_cancers, 2025, doi:10.3390/cancers17010113_

Round 1
Reviewer 1 Report
Comments and Suggestions for Authors
The authors evaluates the prognostic potential of tumor-infiltrating B-lymphocytes in oral cavity squamous cell carcinoma in stages I-IV. The present study included CD19 as a biomarker for B-cells, including a percentage of tumor-infiltrating plasma cells, which has not been studied in OCSCC. This results showed a survival benefit, higher in OCSCC in stages I-II compared to stages III-IV. This results required further research in order to validate prognosis marker, however, this article shows favorable advances for immunotherapy treatments.
The authors may expand the las parragraph of the introduction.
Author Response
Comments: The authors evaluates the prognostic potential of tumor-infiltrating B-lymphocytes in oral cavity squamous cell carcinoma in stages I-IV. The present study included CD19 as a biomarker for B-cells, including a percentage of tumor-infiltrating plasma cells, which has not been studied in OCSCC. This results showed a survival benefit, higher in OCSCC in stages I-II compared to stages III-IV. This results required further research in order to validate prognosis marker, however, this article shows favorable advances for immunotherapy treatments.
The authors may expand the last paragraph of the introduction.
Response: thank you for pointing this out. We agree with this comment. Therefore, we have expanded the last paragraph of the introduction
Reviewer 2 Report
Comments and Suggestions for Authors
This manuscript investigated the prognostic impact of TIL-Bs in early and advanced stage oral cavity squamous cell carcinoma (OSCC). The authors demonstrated that oral cavity squamous cell carcinomas with high levels of CD19+ tumor-infiltrating B-lymphocytes have a more favorable prognosis, irrespective of the total amount of CD45+ tumor-infiltrating lymphocytes and that particularly in early stage disease this could be used to stratify patients who do well on standard-of-care therapy and patients who might need treatment intensification such as adjuvant radiotherapy or pre-operative immunotherapy regimens. This is an interesting observation may provide a potential prognosis for OSCC. However, I have the following suggestions to the authors with further revision for the manuscript.
1. The authors quantified tumor-infiltrating lymphocytes using the percentages (P3 line111-114. 2.3 section). However, pathologically, there are cases with widely varying numbers of lymphocytes infiltrating into the tumor stroma. What is the opinion of this study in this regard?
2. The authors divided “TIL-rich” and “TIL-poor”and so on according to the median percentages. Is this division appropriate? Please show the basis for this division.
3. The authors used “Cohesive” and “Non-cohesive” as tumor pattern of invasion. However, this criterion is subjective, so the authors should use WPOI, YK classification, INF, etc.
Author Response
Comments: This manuscript investigated the prognostic impact of TIL-Bs in early and advanced stage oral cavity squamous cell carcinoma (OSCC). The authors demonstrated that oral cavity squamous cell carcinomas with high levels of CD19+ tumor-infiltrating B-lymphocytes have a more favorable prognosis, irrespective of the total amount of CD45+ tumor-infiltrating lymphocytes and that particularly in early stage disease this could be used to stratify patients who do well on standard-of-care therapy and patients who might need treatment intensification such as adjuvant radiotherapy or pre-operative immunotherapy regimens. This is an interesting observation may provide a potential prognosis for OSCC. However, I have the following suggestions to the authors with further revision for the manuscript.
- The authors quantified tumor-infiltrating lymphocytes using the percentages (P3 line111-114. 2.3 section). However, pathologically, there are cases with widely varying numbers of lymphocytes infiltrating into the tumor stroma. What is the opinion of this study in this regard?
Response: thank you for pointing this out. Indeed, tumors show varying numbers of lymphocytes infiltrating the tumor stroma for each HE-stained slide. Unfortunately, it is technically not feasible to section, stain and analyze each FFPE-block for each tumor, since some resection specimens are made up of dozens of FFPE-blocks. We therefore decided to select the one FFPE-blocks that contained the largest tumor area plus a cuff of normal stroma on the corresponding HE-slide. Since we did this uniformly for each tumor, we believe that the selected FFPE-blocks are equally representative for the corresponding tumor. Moreover, rather than staining a tumor core biopsy, where often tumor stroma is missing, we purposely performed this study on resection specimens, to allow the selection of a tumor area that contains the stromal areas, as well as tumor fields and a tumor-invasive margin. We have added these considerations in our revised manuscript.
- The authors divided “TIL-rich” and “TIL-poor” and so on according to the median percentages. Is this division appropriate? Please show the basis for this division.
Response: thank you for the comment. We did not point this out in the manuscript, but we did perform formal analysis to determine the optimal cut-off point, please see the graphs below. For CD19, the ideal cut-off point would be 0.14%. However, this would leave us with very few TIL-B-poor cases, making it impossible to perform reliable statistics. To ensure that we had enough cases in both the TIL-B-poor group and the TIL-B-rich group for reliable statistics, we decided to go with the median. Since the median of 0.90% (red dotted line) also fell nicely just after the first peak within the distribution (top graph) and within the higher statistical rank (bottom graph), made us confident that this was a reliable cut-off to work with. This has now been included in the revised manuscript both in the text and as Supplementary Figure 2.
- The authors used “Cohesive” and “Non-cohesive” as tumor pattern of invasion. However, this criterion is subjective, so the authors should use WPOI, YK classification, INF, etc.

Reviewer 3 Report
Comments and Suggestions for Authors
In this study, the quantitative results of lymphocytes in tumor tissue are the most important data, but the specific measurement method is not shown. In order for a third party to be able to verify this study, the sampling and measurement methods should be shown. Please describe how the sections were selected from each case, how many sections were used, and which areas on the specimen were counted for cell counts.
For the 222 cases used in this study, it is necessary to explain what criteria were used to select the tissue samples from each case. It is necessary to show that the histopathological sections used for evaluation reflect the typical histological image of the lesion, or to prepare tissue samples from the entire specimen and perform the evaluation.
In the main text, it says, “With this software, the tumor area on each section could be identified, and the percentages of CD45+ and CD19+ immune cells, and CD3+ and CD8+ T-lymphocytes”, but what exactly is the tumor area? Is it the whole of the specimen, or just a part of it? If it is a part of the specimen, the author need to show the criteria for selecting that part.
The percentage of lymphocytes in the results of the immunostaining is not indicated as to what the denominator was. Please indicate what the denominator was.
Author Response
Comments: In this study, the quantitative results of lymphocytes in tumor tissue are the most important data, but the specific measurement method is not shown. In order for a third party to be able to verify this study, the sampling and measurement methods should be shown. Please describe how the sections were selected from each case, how many sections were used, and which areas on the specimen were counted for cell counts.
For the 222 cases used in this study, it is necessary to explain what criteria were used to select the tissue samples from each case. It is necessary to show that the histopathological sections used for evaluation reflect the typical histological image of the lesion, or to prepare tissue samples from the entire specimen and perform the evaluation.
In the main text, it says, “With this software, the tumor area on each section could be identified, and the percentages of CD45+ and CD19+ immune cells, and CD3+ and CD8+ T-lymphocytes”, but what exactly is the tumor area? Is it the whole of the specimen, or just a part of it? If it is a part of the specimen, the author need to show the criteria for selecting that part.
The percentage of lymphocytes in the results of the immunostaining is not indicated as to what the denominator was. Please indicate what the denominator was.
Response: thank you for pointing this out. Please find below the answers to all your questions:
Please describe:
- How the sections were selected from each case: we selected the one FFPE-block from the excision specimen that contained the largest tumor area plus a cuff of tumor-free stroma on the corresponding HE-slide.
- How many sections were used: one per tumor.
- Which areas on the specimen were counted for cell counts: the tumor area consisting of the tumor itself and a cuff of healthy, tumor-free stroma. We have added supplementary Figure 1 to the manuscript with an example of this. The selection of the area is depicted in Supplementary Figure 1B.
- What criteria were used to select the tissue samples from each case: we selected the one FFPE-block from the excision specimen that contained the largest tumor area plus a cuff of tumor-free stroma on the corresponding HE-slide (Supplementary Figure 1A).
- It is necessary to show that the histopathological sections used for evaluation reflect the typical histological image of the lesion: see Supplementary Figure 1A for an example. All HE slides were and tumor area selections were discussed with our HNSCC pathologist. Based on this, the area was manually selected in the software.
- Or to prepare tissue samples from the entire specimen and perform the evaluation: this was considered not feasible since this would be too time consuming. Moreover, in order to be able to select a tumor area that contained a healthy tissue cuff allowing us to accurately determine the tumor area with a tumor-invasive border, we could not use the tumor blocks that lacked a healthy tissue cuff. Also, we opted against selecting a tumor FFPE block at the edge of the tumor, as this often contained a very small are of tumor fields, and was deemed less reliable.
- What exactly is the tumor area? Is it the whole of the specimen, or just a part of it? If it is a part of the specimen, the author need to show the criteria for selecting that part: the tumor area consists of the tumor itself and a cuff of healthy, tumor-free stroma from one 3μm slide. For this study, we found it important that the tumor area would contain the tumor fields, the stromal areas within the tumor, as well as a cuff of tumor-free stroma at the tumor-invasive border, which often contains many lymphocytes. (Supplementary Figure 1B).
- The percentage of lymphocytes in the results of the immunostaining is not indicated as to what the denominator was. Please indicate what the denominator was: the denominator was the tumor area (consisting of the tumor itself and a cuff of healthy, tumor-free stroma).So we determined the % lymphocytes out of the total tumor area. Any white areas within the specimen that contained no cells were excluded from the total tumor area.
This information has now been more extensively explained in the Material and Methods-section, and supplementary figures with examples have now been added. Supplementary Figure 1A represents an HE-slide from the corresponding FFPE-block that was sectioned for IHC-staining. Supplementary Figure 1B shows the delineation of the tumor area used for cell counts.

Round 2
Reviewer 3 Report
Comments and Suggestions for Authors
Appropriate revisions have been made to the parts pointed out by the reviewers.